# Research Advances and Prospects on Mechanism of Sinomenin on Histamine Release and the Binding to Histamine Receptors

**DOI:** 10.3390/ijms20010070

**Published:** 2018-12-24

**Authors:** Yu-Shi Zhang, Jia-Yin Han, Omer Iqbal, Ai-Hua Liang

**Affiliations:** 1Key Laboratory of Beijing for Identification and Safety Evaluation of Chinese Medicine, Institute of Chinese Materia Medica, China Academy of Chinese Medical Sciences, Beijing 100700, China; yszhang@icmm.ac.cn (Y.-S.Z.); jiayints@163.com (J.-Y.H.); 2Stritch School of Medicine, Loyola University Chicago, Chicago, IL 60153, USA; oiqbal@luc.edu

**Keywords:** sinomenine, pharmacological mechanism, adverse reaction mechanism, clinical application, histamine/histamine receptor, virtual molecular docking

## Abstract

Sinomenine (SIN) is widely used in China to treat a variety of rheumatic diseases (RA), and has various pharmacological effects such as anti-inflammatory, analgesic, and anti-tumor effects. However, due to the histamine release characteristics of SIN, its adverse reactions such as allergic reactions, gastrointestinal reactions, and circulatory systemic reactions have been drawing increasing attention. We present here a systematic review of the chemical structure, pharmacological effects, clinical application, and adverse reactions of SIN, a detailed discussion on the relationship between histamine/histamine receptor and mechanism of action of SIN. In addition, we simulated the binding of SIN to four histamine receptors by using a virtual molecular docking method and found that the bonding intensity between SIN and receptors varied in the order shown as follows: H1R > H2R ~ H3R > H4R. The docking results suggested that SIN might exhibit dual regulatory effects in many processes such as cyclooxygenase-2 (COX-2) expression, NF-κB pathway activation, and degranulation of mast cells to release histamine, thereby exhibiting pro-inflammatory (adverse reactions)/anti-inflammatory effects. This study provides a theoretical basis for the clinical treatment of inflammations seen such as in RA using SIN, and also suggests that SIN has great potential in the field of cancer treatment and will have very important social and economic significance.

## 1. Introduction

Sinomenine (SIN) is an alkaloid monomer extracted from the medicinal rhizome of *Sinomenium acutum*, a plant of the family Menispermaceae used in traditional Chinese Medicine (TCM) [1]. *Sinomenium acutum* has been used for medicinal purpose in China for over 2000 years, and it has been recorded in ancient TCM books such as *Shen Nung Pen Ts’ao* and *Chin k’uei Yao Lueh*. Originally extracted from Japanese *Sinomenium acutum* in the 1920s by Ishiwari [2], SIN is a strong histamine-releasing agent with a variety of pharmacological effects such as anti-inflammatory, immunosuppressive, analgesic, antihypertensive, and anti-arrhythmic effects [3,4]. The dosage forms of SIN have been used for clinical treatment of rheumatoid arthritis (RA) in China and Japan [2], which include injection, enteric-coated tablets, and sustained-release tablets [5]. In recent years, the anti-tumor effect of SIN has drawn worldwide attention [6,7,8]. However, SIN also has some noticeable characteristics, which include a relatively short biological half-life, the potential of causing clinical adverse reactions such as rash and gastrointestinal reactions by promoting histamine release, and being unstable and easily decomposable under acid, alkali, light, or heat conditions [9]. Here, we systematically summarized the structure, pharmacological effects, adverse reactions and the mechanism of histamine release in order to clarify the pharmacological mechanisms of SIN and further promote its clinical application.

### Chemical Structure of SIN

The chemical name of SIN is 7,8-didehydro-4-hydroxy-3,7-dimethoxy-17-methylmorphinan-6-one, and its molecular formula is C_19_H_23_NO_4_ with a 329.38 relative molecular mass [10]. With a chemical structure (Figure 1a) similar to opioids such as morphine (Figure 1b) and codeine (Figure 1c) [11], it consists of a hydrogen phenanthrene core and ethylamine bridge. Therefore, SIN possesses certain pharmacological properties similar to those of morphine and related compounds.

The SIN skeleton is mainly composed of four rings: A, B, C, and D (Figure 1a), in which ring A is a benzene ring, ring B is a half-chair hexatomic ring connected to ring A, ring C is a twisted-chair hexatomic ring possessing a conjugated unsaturated ketone structure connected with ring B, and ring D is a nitrogen-containing hexatomic ring [12]. Unlike morphine, the epoxidation bridge of SIN is broken, and a phenolic hydroxyl group is present at its position four in the molecular structure while it is at position three for morphine [11].

The structural modification and optimization of SIN has been an increasingly high-profile concern due to its potential to cause adverse reactions, and instability of physicochemical properties. In order to identify its structure and study the chemical properties, Japanese scholar Manske synthesized some SIN derivatives, but no biological activity was reported. Existing studies primarily have focused on the structural modification of active groups on the four rings of SIN. The structurally-modifiable sites in SIN A ring include 1-position atom H and 4-position hydroxyl group. The structural modification of B-ring is performed mainly at positions 10 and 14, and there are three chemically active sites in ring C, including the carboxyl (C-6), methoxyl (C-7), double bond with C-7 and C-8, and the structural modification of ring D is performed mainly on atom N [13,14,15]. Jian Tang [16] et al. presented a systematic review of the modification and biological activity of over 500 SIN derivatives and obtained many novel morphine-type anti-inflammatory and immunosuppressive agents. The findings showed that 1-substituted, 4-OH ether and ester and pyrazine-fused ring remained as an effective strategy for designing SIN derivatives with better therapeutic effects.

In addition, alkaloid drugs in various structures are metabolized through complex pathways, generating many products. At present, the SIN metabolites have not been profoundly studied, and the molecular structure and pharmacological activity of metabolites are yet to be further analyzed.

## 2. Pharmacological Effects of SIN

### 2.1. Immunosuppressive and Anti-Inflammatory Activities

RA is a chronic systemic inflammatory disease [17], and it is treated with SIN based on the following main mechanisms: Inhibiting the proliferation of inherent synovial cells and promoting their apoptosis; suppressing the activities of immunocytes such as lymphocytes, dendritic cells (DC), and macrophages (MO), as well as inducing a tendency to balance through regulating the functions of the body’s immune system; inhibiting the binding activity of NF-κB by up-regulating the expression of IκB-α, causing a decrease in the levels of inflammatory cytokines TNF-α, IL-1β, etc., thereby relieving the local inflammatory reaction [18,19,20,21]. Furthermore, SIN reduces the synthesis of prostaglandin E2 (PGE2) induced by lipopolysaccharide (LPS)-activated monocytes, through suppressing cyclooxygenase-2 (COX-2) [22]. It can also act on the mitogen-activated protein kinases (MAPKs) pathway to suppress the increase of synthesis of intercellular adhesion molecule-1 (ICAM-1) and nitric oxide (NO) caused by LPS and protect the vascular endothelia after inflammation [23]. Hui-Fang Xiong et al. [24] established mouse models with inflammatory bowel disease in their study and they revealed that SIN suppressed the activation of TLR/NF-κB signaling pathway induced by dextran sulfate sodium salt (DSS), down-regulated the expression of TLR4, MyD88, and NF-κBp65, and promoted the decrease of proinflammatory cytokines, thereby reducing intestinal inflammation.

Therefore, it is evident that, the immunosuppressive and anti-inflammatory activities of SIN are achieved mainly through the regulation of immunocytes (T lymphocytes, B lymphocytes, DC, MO and monocytes), inflammatory cytokines, adhesion molecules, inflammation-related inducible enzyme (COX-2) and signal transduction system (NF-κB, TLR, MAPK pathway).

### 2.2. Analgesic and Sedative Effects

The moderate sedative and analgesic effects of SIN are ascribed to its chemical structure being similar to that of morphine [25]; the site action of SIN is a nerve center, but its mechanism of action remains unclear. Jeong-Yun Lee et al. [26] found that intraplantar application of SIN suppressed formalin-induced pain behavior, and the results demonstrated that SIN had a peripheral analgesic effect by inhibiting I_Na_. Qing Zhu et al. [27] believed that SIN could significantly alleviate the neuralgia induced by chronic constrictive injury and improve depression-like symptoms in rats; moreover, this pain-relieving effect might be blocked by bicuculline, a γ-aminobutyric acid type A (GABA_A_)-specific receptor antagonist, indicating that this analgesic effect of SIN was mediated through the GABA_A_ receptors. Yun-Tao Ou et al. [28] also showed that N-demethylsinomenine, one of the SIN metabolites, exerted behaviorally-specific anti-allodynia against postoperative allodynia mediated through the GABA_A_ receptors.

SIN could significantly suppress animals’ spontaneous and passive activities, and it possessed sedative and tranquilizing effects, with the potential of affecting the central nervous system. SIN exhibited a marked inhibitory and blocking effect on the naloxone-induced withdrawal contracture in morphine-dependent isolated guinea pig ileum, and it exerted a preventive and therapeutic effect on opioid dependence [29]. It could also eliminate the conditioned place preference (CPP) generated by morphine, with an underlying mechanism of action related with the reduction of central cAMP levels and regulate the disorder of neurotransmitter levels in morphine-dependent rats [30]. Si-Wei Chen et al. [31] found that SIN could exhibit anxiolytic-like effects in the elevated plus-maze, light/dark transition and social interaction test models, and the mechanism might lie in the regulation of noradrenaline, dopamine, and 5-HT in the brain.

### 2.3. Anti-Tumor Effects

SIN effectively suppressed the activity of MDA-MB-231 and 4T1 breast cancer cells, and significantly inhibited the invasion and migration capacity of these cells by down-regulating NF-κB activation mediated by IL-4/miR-324-5p/CUEDC2 axis [32]. Further studies revealed that, in a breast cancer-lung metastasis model, SIN inhibited the NF-κB-mediated Sonic hedgehog (SHh) signaling pathway by blocking the NF-κB activation, and then regulated the cell metastasis markers such as MMP-2, vimentin to affect cancer metastasis [33]. Li-Ping Zhou et al. [34] found that activation of PI3K/Akt and MEK/ERK signaling pathways could antagonize SIN-induced lung cancer cell line NCI-H460 apoptosis, and the suppression of these two signaling pathways could increase the SIN-induced apoptosis levels. SIN suppressed proliferation of human cervical cancer Hela cells by inducing them to apoptosis, the mechanism of which may be associated with the P13K/Akt-Caspase 3 signal transduction pathway [35]. It also inhibited the proliferation of human hepatoma cells HepG by down-regulating the expression of COX-2 and VEGF [36]. Fei Deng et al. [37] found that SIN enhanced autophagy in renal cell carcinoma ACHN cells via inactivating PI3K/AKT/mTOR pathway, up-regulated the expression levels of apoptotic proteins (caspase-3 and caspase-9) and finally promoted apoptosis.

SIN suppresses the proliferation of various tumor cells, such as breast cancer, lung cancer, cervical cancer, hepatic cancer, and renal cancer, and its anti-tumor mechanism may involve the following factors: Antagonizing tumor cell invasion and migration; inhibiting tumor cell proliferation via immunosuppression (COX-2 and VEGF) or apoptotic protein-associated pathway (NF-κB, PI3K/AKT) to induce tumor cell apoptosis.

### 2.4. Other Pharmacological Effects

SIN may affect cardiac function and possess obvious antihypertensive and antiarrhythmic effects [38]. Experiments have validated that SIN could prolong the left ventricular ejection time (LVET) and Q-S2 interval (QS2) during rabbits’ heart contraction and inhibit Na^+^ and K^+^ transmembrane current [39]. However, the effects of SIN on cardiac function and its antiarrhythmic mechanism have not been very clearly understood. Studies have also shown that SIN could inhibit the endothelin-stimulated proliferation and DNA synthesis of isolated rabbit vascular smooth muscle cells in a dose-dependent manner and had the effects of antagonizing proliferation of vascular smooth muscle cells [40]. Zhi-Qing Zhao et al. [41] observed the therapeutic effect of SIN on renal ischemia/reperfusion injury. The results supported that SIN relieved the occurrence of edema and tubular necrosis after renal ischemia, reduced the levels of urea nitrogen and creatinine, which had increased after the renal function damage, alleviated the deterioration in renal function and eased the renal tubular cell apoptosis induced by ischemia/reperfusion.

The pharmacological effects of SIN were summarized in Table 1.

## 3. Clinical Application and Adverse Reactions of SIN

### 3.1. Clinical Application

SIN is mainly used in the clinical treatment of diseases such as RA. The current medicine dosage forms include Zhengqing Fengtongning (ZQFTN) tablets and Sinomenine Hydrochloride Injection (SHI). ZQFTN is a Chinese patent medicine made after the refinement of SIN, an active component extracted from the medicinal rhizome of *Sinomenium acutum* in China.

**Treatment of RA:** The possession of anti-inflammatory, analgesic and immunosuppressive action is the pharmacological basis for using SIN in the treatment of RA. Yu-Lian Cheng [42] applied ZQFTN tablet to treat 80 patients of RA, achieving a response rate of 82.6%. Wei-Wei Liu et al. [43] systematically evaluated the efficacy and safety of SIN in treating RA by searching the China National Knowledge Infrastructure (CNKI) Database, Wanfang Database, Pubmed, Cochrane Library and other databases electronically. Sixteen randomized controlled trials (RCTs) were employed, involving a total of 1500 subjects. The results of this meta-analysis indicated that on basis of Western medicine (methotrexate, MTX), SIN was more effective in total effective rate (*p* < 0.001). SIN alone versus MTX also showed advantages in RA therapy (*p* < 0.05). Besides, adverse events occurred less frequently in combination of SIN and MTX than MTX alone (*p* < 0.001).

**Treatment of knee osteoarthritis (KOA):** Shan-Shan Lin et al. [44] systematically evaluated the efficacy and safety of SIN (ZQFTN) in treating KOA by retrieving literature from databases such as PubMed, Cochrane Library, EMbase, and CNKI. Seven RCTs were employed, involving a total of 566 subjects. The results of this meta-analysis showed that, compared to the monotherapy of Western medicine, ZQFTN (combination with a Western medicine) could significantly decrease the arthralgia, ankylosis, and Lequesne index scores in terms of treating KOA (*p* < 0.05) and reduce the serum TNF-α and IL-1β concentrations (*p* < 0.05), with a higher safety profile.

**Treatment of ankylosing spondylitis (AS):** SIN (ZQFTN) was applied to treat 67 patients with AS. After one-week of medication, the pain in most patients was significantly reduced, indicating that SIN could control clinical symptoms and lower the active inflammation indicators in a short period of time, so it turned out to be remarkably effective in relieving arthralgia and arthrocele [45]. Shan-Hao Liang [46] systematically evaluated the efficacy and safety of SIN (ZQFTN) in treating AS by retrieving literature from PubMed, EMBASE-ASP and other databases. 12 RCTs were employed, involving 1112 patients. Meta-analysis results showed that the administration of ZQFTN would improve overall response rate, spinal pain, and generalized pain.

**Treatment of glomerular disease:** Findings displayed that the treatment of glomerular diseases with SIN not only reduced the levels of protein in the urine but also alleviated the hematuria symptoms; moreover, the incidence rate of adverse effects was significantly lower than that of the commonly used Tripterygium glycosides tablets in clinical practice [47]. In addition to the capacity of reducing the excretion of urine protein, SIN could also decrease hematuria, with a lower incidence rate of adverse effects than that of Tripterygium glycosides tablets [48].

### 3.2. Adverse Reactions

With the wide clinical application of SIN, its adverse reactions such as allergic reactions, digestive system, blood system, circulatory system, and nervous system damage have been drawing increasing attention [19].

Zhao-Zhao Wu et al. [49] reported that of the 4064 patients in seven hospitals who took ZQFTN sustained-release tablets, 193 experienced adverse drug reactions, with an incidence of adverse reactions of 4.75%. The adverse reactions principally involved the skin and digestive system, and the main clinical manifestations included allergic reactions and gastrointestinal reactions such as rash, itching, dizziness, headache, nausea, and vomiting, and decreased appetite.

In addition, Jue-Rong Chen et al. [50] reported the agranulocytosis which occurred after the use of SIN to treat 44 patients with systemic lupus erythematosus and systemic sclerosis. Then the drug was withdrawn and replaced by granulocyte colony-stimulating factor and antibiotics to manage the adverse reactions. The count of granulocytes returned to normal 10 days thereafter. This suggested that it was necessary to closely monitor bone marrow function and granulocyte count during the SIN treatment.

## 4. Mechanism of Action of SIN and Its Relationship with Histamine/Histamine Receptor

Hiroshi, a Japanese scholar, first discovered that SIN was a strong histamine releasing agent [51]. Histamine is a major mediator of allergic diseases with a wide range of effects on many cell types, and it is mediated by specific surface receptors on target cells [52]. The histamine receptors in humans can be divided into four receptors, H1, H2, H3, and H4. These receptors are G-protein coupled receptors (GPCRs), which convert extracellular signals into intracellular second signal systems through G proteins, but their coding sites, expression, signal transduction, and function vary [53]. H1R is mainly distributed on the surface of vascular endothelial cells, smooth muscle cells, neurons, and immunocytes of skin and mucosae. Histamine regulates body function via H1R, resulting in vasodilation, increased vascular permeability, decreased blood pressure, headache, tachycardia, bronchial spasm and other effects. H2R is found in a variety of tissues and cells, including the brain, gastric parietal cells, smooth muscle cells, and is responsible for relaxation of smooth muscle cells in the blood vessels, uterus, and airways. H3R is principally expressed on the surface of histaminergic neurons and its function is mainly to regulate the release of neurotransmitters such as histamine and acetylcholine, subsequently initiating a negative-feedback regulation of the histamine synthesis and release. H4R is a newly discovered histamine receptor that is highly expressed in tissues and hematopoietic cells associated with inflammatory responses [54,55].

### 4.1. The Mechanism of Histamine Release in Adverse Reactions of SIN

SIN can rapidly cause marked swelling and degranulation of mast cells in rats’ ascitic fluid, then histamine is removed from mast cells. A previous study has shown that SIN promoted the phosphorylation of extracellular signal-regulated kinase (ERK), the cleavage of Annexin A1 (ANXA1), cytosolic phospholipase A2 (cPLA2) phosphorylation, as well as COX-2 expression in a dose-dependent manner, ultimately leading to degranulation of rat basophilic leukemia 2H3 (RBL-2H3) cells. PD98059 (MEK inhibitor) prevented ERK phosphorylation by binding to ERK-specific MAP kinase MEK, indicating that SIN mediated the degranulation of RBL-2H3 cells to release histamine by up-regulating ERK phosphorylation levels [56]. Likewise, SIN induced the degranulation of P815 cells to release inositol-1,4,5-trisphosphate (IP3) and histamine in a dose-dependent manner in spite of the differences in mechanism of action. The binding of SIN-specific IgE to its FcεRI receptor on mast cells triggered aggregation and activation of Lyn, which resulted in the phosphorylation of Syk tyrosine kinase and led to the activation of the mast cell. After the IgE-mediated mast cell activation, numerous signaling proteins were further phosphorylated, then triggering the PLCγ-mediated IP3 production. The cytoplasmic free IP3 was bound to the endoplasmic reticulum IP3R, triggering Ca^2+^ release, which resulted in the degranulation of P815 cells and the release of the anaphylactic mediator histamine. In addition to the IgE-mediated route, mast cells could be activated in vivo as shown in investigations on the allergic effect of SIN in mice. SIN could enhance blood vessel permeability and dose-dependently increase IP3 and TNF-α levels in serum. Therefore, the in vitro and in vivo allergic effects of SIN in mast cells were confirmed [57]. Further studies have shown that SIN could interact with Mas-related G protein-coupled receptor X2 (MRGPRX2) to induce PLCc1 and IP3R phosphorylation in mast cells. Then it directly promoted diacylglycerol and IP3 synthesis via the MRGPRX2-PLC pathway, causing the release of Ca^2+^ from the endoplasmic reticulum, which ultimately led to the degranulation of mast cells to release β-hexosaminidase, histamine, and TNF-α as well as chemokines, such as MCP-1, IL-8, and MIP-1b. This illustrated that the mast cell-specific receptor MRGPRX2 might serve as a receptor for SIN to trigger mast cell degranulation—a key process in anaphylactic reactions [58].

Histamine release increased vascular permeability and induced the relaxation of vascular smooth muscle, thus it remained as a key substance for SIN-induced adverse reactions, especially allergic reactions. Lu-Fen Huang et al. [59], by using two animal models, first demonstrated that SIN induced a typical allergic reaction in vivo, and relieved SIN-induced allergic reactions with mast cell membrane stabilizers and H1R antagonists. The results showed that SIN-induced allergic reactions were associated with activation of NF-κB signaling. SIN could cause IκBα degradation and phosphorylation, and NF-κB was released and translocated into the nucleus, producing pro-inflammatory cytokines, which then triggered a series of adverse reactions.

### 4.2. The Mechanism of Histamine and Its Receptors in Pharmacological Effects of SIN

Mast cells are not only the main effector cells of hypersensitivity and inflammation but also play an important role in many chronic inflammatory diseases and innate immunity. It is well known that SIN at a high concentration can directly stimulate degranulation of mast cells. Wen-Jun Wang et al. [60] found that SIN at a high concentration (>500 μmol/L) could directly induce degranulation of RBL-2H3 cells, while SIN inhibited the proliferation of RBL-2H3 cells and promoted apoptosis at a low concentration (100 μmol/L), which showed marked inhibition on antigen-activated RBL-2H3 cell degranulation. This result showed that SIN initially depleted the mast cells, causing their degranulation, to release inflammatory mediators such as histamine, and then inhibited the degranulation of mast cells with activated immune function. It was evident that SIN played a two-way regulation on mast cell degranulation. The potential mechanism of molecular regulation needs to be further studied.

Histamine has a certain connection with the analgesic effect. The combined treatment of antihistamine diphenhydramine or promethazine with SIN may produce a marked synergistic effect in relieving pain. Ming-Fa Zhang [61] claimed that the SIN-released histamine might weaken its inherent analgesic effect, and the analgesic effect reached the peak with the exhaustion of histamine, along with the gradual depletion of histamine when the medication was continued. This conclusion was consistent with a clinical report saying that the analgesic action became enhanced as patients continued the medication.

Studies have revealed that SIN (30 or 100 μmol/L) could significantly reduce the extra high level of cAMP in morphine-dependent cells during withdrawal, and additionally regulate the cGMP level, thus it exhibited inhibitory action on naloxone-induced withdrawal symptoms in morphine-dependent rats and CPP in mice [62]. However, high-dose SIN (300 μmol/L) may cause a significant increase in intracellular cAMP level [63]. These results indicated that SIN might exert a dual action, both inhibition and facilitation of the central histamine system, but the mechanism of action remained unclear.

Zhi-Xian Mo et al. [64] performed an experiment on isolated guinea pig ileum to observe the histamine-induced tension changes in ileal smooth muscle contraction. The results showed that SIN exhibited dual effects on histamine in vivo: On one hand, it might induce mast cell to release histamine; on the other hand, it inhibited the ileal contractile response caused by histamine. In view of the dual and opposite effects of SIN on histamine, the mechanism of SIN was subject to the influence of body functional state, thereby exhibiting different effects.

Based on the dual mechanism described in the above studies, we assumed that SIN might induce tolerance effect with the prolonged time of action, and its binding to histamine receptors, which might exert certain antagonistic or inverse agonistic effects on histamine. Therefore, we simulated the binding of SIN to the four histamine receptors by using a virtual molecular docking method, and then we further explored the mechanisms of histamine on the pharmacological effects of SIN.

### 4.3. The Docking of SIN with Histamine Receptors H1R, H2R, H3R, H4R

Histamine receptor belongs to the GPCR family and has seven transmembrane structures (TM1–TM7) with a total of 487 amino acids. Homology modelling was performed on H2R, H3R, and H4R by using the swiss-model software [65,66] with the H1R transmembrane region as a template. The structure after modeling was shown in Figure 2a,b. The sequence comparison with amino acid residues in the H1R, H2R, H3R, and H4R binding cavities showed that TM3, TM6, and TM7 were relatively highly conserved regions, TM5 domain was a variable region, and the key binding sites were distributed in TM3, TM5, and TM6 domains (Figure 3). By the selected ligand 5EH in PDB:3RZE and with the amino acid residues within the 10A range of protein active region as active sites, the receptors were molecularly docked using GOLD5.3. The docking scores between SIN and four target proteins H1R, H2R, H3R, and H4R were 85.99, 40.75, 41.23, and 35.81, respectively. As the binding modes shown in Figure 2c, H1R got the highest score, H2R has a consistent binding mode to H1R, while H3R and H4R have a similar binding mode.

The binding mode of SIN and H1R was shown in Figure 4a,b. SIN formed a π–π conjugated bond by acting with Phe435 and Phe432 in the TM6 domain and Tyr108 in the TM3 domain, and a π-sigma conjugated bond with Trp428 in the TM6 domain. These interactions enhanced the binding intensity between SIN and H1R. SIN also generated hydrogen-bond interactions with the Ser111 and Thr112 in the TM3 domain and Thr194 in the TM5 domains. Therefore, compared with the other three receptors, H1R generated more hydrophobic interactions and hydrogen-bond interactions, therefore, it had better binding intensity than the other three receptors. Compared to the binding mode of H1R, the Val99 in TM3 domain of H2R substituted the Tyr108, losing the π–π conjugated bond interaction of the benzene ring. Similarly, Cys102 substituted Ser111, losing the hydrogen-bond interaction, which greatly attenuated the interactions between SIN and H2R (Figure 4c,d). Compared with H1R, H1R-Phe435 and H1R-Phe432 in TM6 domain of H3R were turned into H3R-Met378 and H3R-Thr375, which changed the spatial structure of TM6 domain and weakened hydrophobic interaction, thus the binding mode of SIN and H3R was reversed. SIN bound only with H3R via the Ser203 and Glu206 in the TM5 domain and Thr199 in the TM3 domain, forming relatively weak hydrogen-bond interaction (Figure 4e,f). As shown in Figure 4g,h, the binding mode between SIN and H4R was similar to H3R, the conjugated bond was likewise missing in the TM6 domain. Since the H3R-Ala122 in the TM3 domain was turned into H4R-Val102, the absence of hydrogen-bond interaction of H4R resulted in a change in spatial conformation, which lead to the lowest score of H4R. In brief, the bonding intensity between SIN and the four histamine receptors varied in the order shown as follows: H1R > H2R ~ H3R > H4R.

We found that in the mechanisms underlying adverse reactions, SIN up-regulated COX-2 expression [56], activated NF-κB pathway [59], and potentiated the degranulation of mast cells to release histamine; but this seemed to be contradictory with the anti-inflammatory mechanisms, in which SIN could down-regulate COX-2 expression [22] and inhibit the NF-κB [24] signaling pathway (Figure 5). The conclusion on virtual molecular docking indicated that SIN bound to histamine receptors, especially with higher efficiency of binding to H1R. It is enough to explain that SIN, upon entering an organism, up-regulates the COX-2 expression, activates the NF-κB pathway, and promotes the degranulation of mast cells to release large amounts of histamine, then there is a relatively low likelihood of causing a series of inflammations and adverse reactions. As the action proceeds, SIN binds to histamine receptors and reduces the histamine-to-receptor binding, thereby down-regulating COX-2 levels, inhibiting the NF-κB pathway, and exerting anti-inflammatory effects. Therefore, SIN would exhibit dual regulatory effects in using this process, but more experimental studies are still needed to support the detailed mechanism of action.

## 5. Prospects

*Sinomenium acutum* is a traditional therapeutic drug in both China and Japan, and the research on the pharmacological activity of *Sinomenium acutum* is mainly focused on its active ingredient SIN at present. In recent years, numerous scholars have performed a large number of experimental studies and validated that SIN had various effects such as immunosuppression, anti-inflammation, analgesia, and apoptosis induction. The comprehensive discussion of the mechanism of action of SIN and histamine provides a theoretical basis for the clinical treatment of inflammatory diseases such as RA, and also presents data support as early as possible for the application of SIN in the field of organ transplantation or developing it as a novel immunosuppressant. More importantly, SIN is an effective anti-tumor component with multiple anti-tumor effects such as suppressing tumor cell proliferation, inducing tumor cell apoptosis, and inhibiting tumor cell invasion and migration, and its mode of action is characterized by multiple targets and approaches. In addition, SIN may exert synergistic effects when used in combination with multiple chemotherapeutics drugs, such as 5-fluorouracil [67,68], cisplatin [69], and carboplatin [70]. It is evident that SIN also has good prospect and application potential in the field of clinical treatment of cancer. Further studies should focus on the exploration of the mechanism of action of the anti-tumor effect of SIN at the molecular/genetic level. Further extensive investigation of the relationship between experiment and clinical treatment based on the anti-tumor effect of SIN to provide a scientific basis for the development of innovative anti-cancer drugs, will be of very important social and economic significance.

However, the clinical adverse effects such as allergic reactions and gastrointestinal reactions caused by SIN through histamine release have severely impeded the further clinical application of SIN. For people with allergic constitution, it should be taken in small doses and be used sparingly or forbidden. During the administration period, people should avoid taking high-fat and high-protein diets. For digestive tract reactions, the irritation and instability of SIN can be altered only by selecting appropriate dosage forms. However, the experimental study on transdermal administration of SIN is still on-going and its clinical application has not been reported, therefore further studies are still needed. The strong histamine release action of SIN weakens its analgesic effect and remains a major cause of allergic and gastrointestinal reactions. The combination of SIN with antihistamines can not only enhance its analgesic effect but also antagonize its stimulating effects on the gastrointestinal tract and prevent allergic reactions.

We did not find information about the antagonistic activity of SIN, despite of the great efforts that had been made in searching an extensive range of literature. The results of the virtual molecular docking of SIN have shown its potential to bind to histamine receptors. At present, no experimental study has yet been found, thus it is meaningful to investigate the histamine antagonistic activity of SIN in future studies.

Although the studies on the pharmacological effects and adverse reaction mechanism of SIN have reached the level of cellular molecules, those at the level of gene and receptor are still insufficient due to the numerous signal transduction pathways involved. SIN will have a broader application prospect in clinical treatment if its molecular biological mechanism is further clarified.

## Figures and Tables

**Figure 1 ijms-20-00070-f001:**
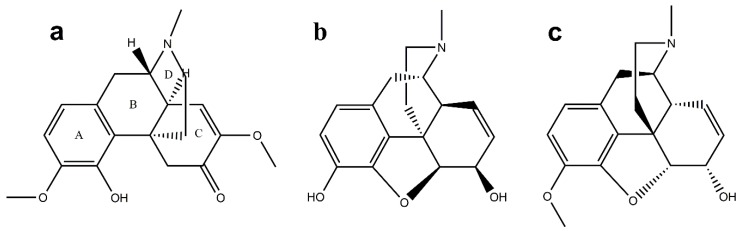
Stereochemistry structure of sinomenine (**a**), morphine (**b**) and codeine (**c**).

**Figure 2 ijms-20-00070-f002:**
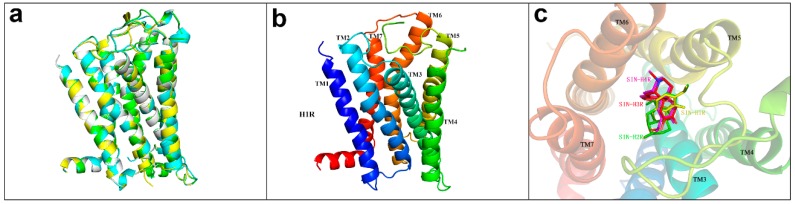
(**a**) Superimposed charts of histamine receptor protein structure, with H1R-white, H2R-green, H3R-blue, and H4R-yellow; (**b**) 7 transmembrane domains of H1R, with TM1-blue, TM2-marine, TM3-cyan, TM4-green, TM5-yellow, TM6-orange, TM7-red; (**c**) binding mode diagram of SIN and four target proteins using ball-and-stick model: H1R-yellow, H2R-green, H3R-red, H4R-magenta.

**Figure 3 ijms-20-00070-f003:**
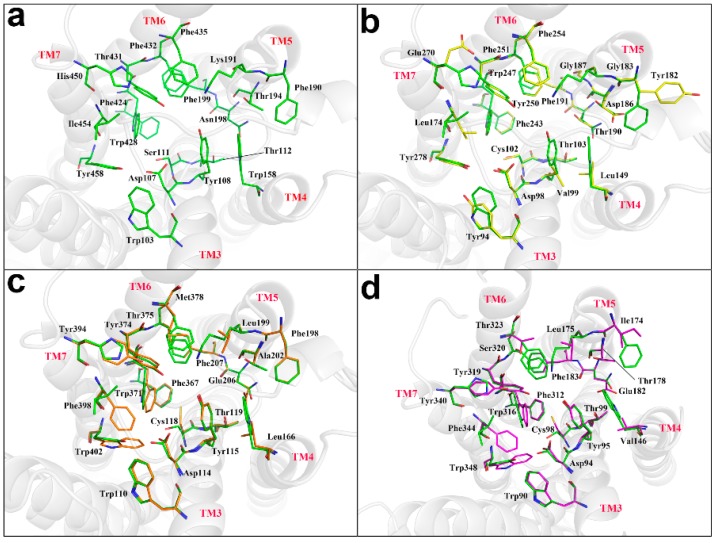
Schematic diagram of the histamine receptor structure binding cavity, with 7 transmembrane domains labeled by the gray cartoon. (**a**) The H1R cavity binding site was shown by a green ball-and-stick model; (**b**) superimposed charts of H1R and H2R, with H1R-green, H2R-yellow; (**c**) superimposed charts of H1R and H3R, with H1R-green and H3R-orange; (**d**) superimposed charts of H1R and H4R, with H1R-green and H4R-magenta.

**Figure 4 ijms-20-00070-f004:**
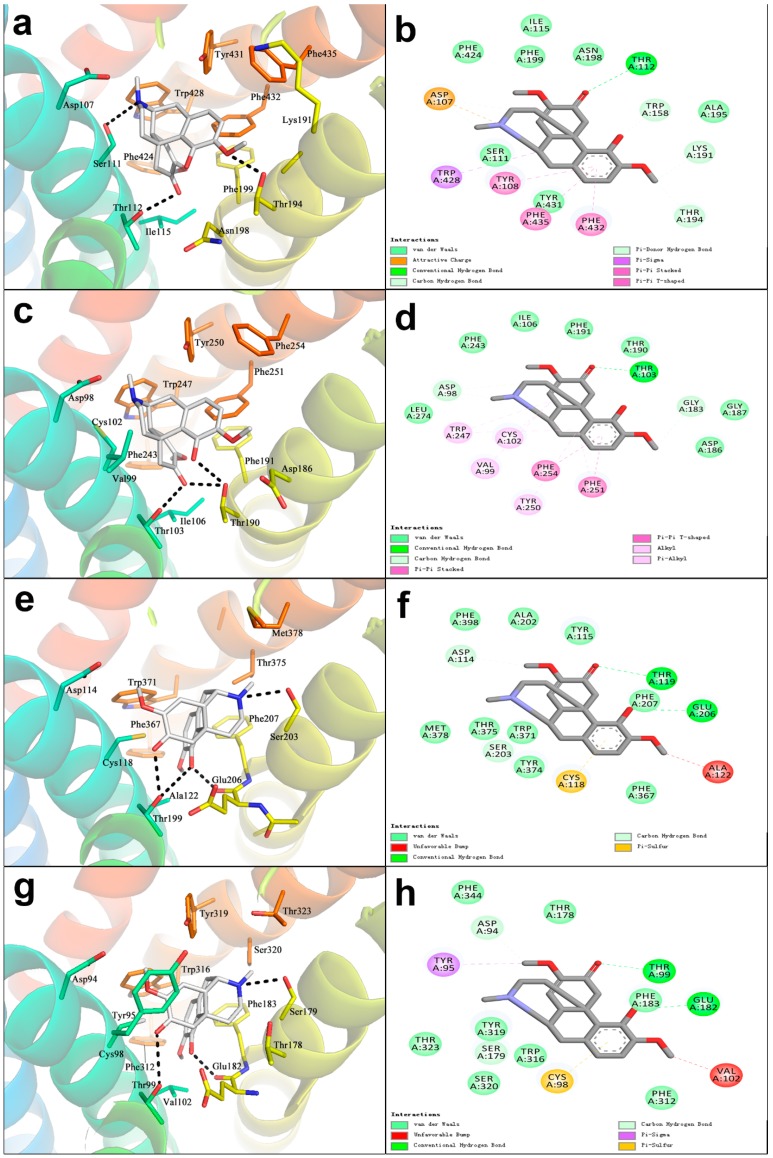
Binding mode of SIN and histamine receptors. (**a**) Conformational binding mode of H1R and SIN after docking, where: The gray ball-and-stick model represented the conformation of post-docking SIN and the interactions of amino acid residues, and black lines marked the hydrogen-bond; (**b**) two-dimensional binding mode diagram of H1R-SIN; (**c**) conformational binding mode of H2R and SIN after docking; (**d**) two-dimensional binding mode diagram of H2R-SIN; (**e**) conformational binding mode of H3R and SIN after docking; (**f**) two-dimensional binding mode diagram of H3R-SIN; (**g**) conformational binding mode of H4R and SIN after docking; (**h**) two-dimensional binding mode diagram of H4R-SIN.

**Figure 5 ijms-20-00070-f005:**
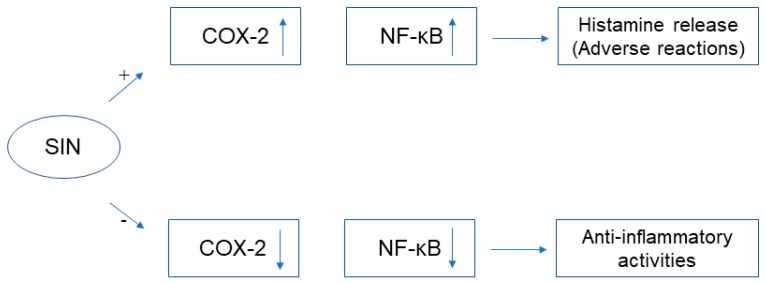
The dual regulation mechanism of SIN, (+) indicated that SIN could up-regulate (↑) cyclooxygenase-2 (COX-2) expression, activate (↑) NF-κB pathway and promote mast cell degranulation to release histamine, further causing adverse reactions; (−) indicated that SIN could down-regulate (↓) COX-2 expression and inhibit (↓) NF-κB signaling pathway, thus exerting anti-inflammatory effects.

**Table 1 ijms-20-00070-t001:** The pharmacological effects of sinomenine (SIN) along with citations.

Pharmacological Effects of SIN	References
Anti-inflammatory activities	[17,18,19,20,21,22,23,24]
Analgesic effects	[25,26,27,28]
Sedative effects	[29,30,31]
Anti-tumor effects	[32,33,34,35,36,37]
Antiarrhythmic effects	[38,39]
Other pharmacological effects	[40,41]

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
