# Peer review of "Research Advances and Prospects on Mechanism of Sinomenin on Histamine Release and the Binding to Histamine Receptors"

_ijms, 2018, doi:10.3390/ijms20010070_

Round 1
Reviewer 1 Report
This review is addressed to deepen knowledge on the mechanism of histamine release by Sinomenin in a very satisfactory way. The work is interesting, informative and well-written, and I really enjoyed reading it. I only have some observations that are listed in the following lines.
1. Page 2: Summarize the pharmacological effects of SIN along with citations in a table.
2. Page 5: Provide a short graphical presentation that summarizes the mechanism.
Author Response
Dear Reviewer,
I am very grateful that you can review this manuscript.
Response to Reviewer 1 Comments.
Point 1: Page 2: Summarize the pharmacological effects of SIN along with citations in a table.
Response 1: The pharmacological effects of SIN were summarized in Table 1.
Point 2: Page 5: Provide a short graphical presentation that summarizes the mechanism.
Response 2: Figure 5. The dual regulation mechanism of SIN, (+) indicated that SIN could up-regulate COX-2 expression, activate NF-κB pathway and promote mast cell degranulation to release histamine, further causing adverse reactions; (-) indicated that SIN could down-regulate COX-2 expression and inhibit NF-κB signaling pathway, thus exerting anti-inflammatory effects.
Thank you for your suggestion.
Best regards.
Reviewer 2 Report
Liang summarized the recent development of Sinomenin. The authors systematically summarized the structure, pharmacological effects, adverse reactions and the mechanism of histamine release of SIN which may get a bunch of attention from the medicinal chemists, natural product researchers and pharmaceutical scientists. The presentation here is pretty good, but still, has something really confused me.
The authors stated in line 41~44 "SIN has its associated disadvantages". If the SIN has so many disadvantages why scientists are still interested in SIN. What are the advantages of SIN? So I suggest the authors reorganize their introduction section.
Line 39, 40, 41 the sentence needs to be revised to be scientific writing.
One suggestion for the authors is that please show the stereochemistry of the structures in Figure 1.
Author Response
Dear Reviewer,
I am very grateful that you can review this manuscript.
Response to Reviewer 2 Comments.
Point 1: The authors stated in line 41~44 "SIN has its associated disadvantages". If the SIN has so many disadvantages why scientists are still interested in SIN. What are the advantages of SIN? So I suggest the authors reorganize their introduction section.
Response 1: In the first half of the introduction section, we showed that SIN has anti-inflammatory, immunosuppressive, analgesic and anti-tumor pharmacological effects, and it is widely used in clinical treatment. These are the advantages of SIN and the research focus of many scientists. Although SIN has disadvantages such as adverse reactions, exploring how to alleviate adverse reactions through the mechanism studies are also the direction of interest of many scientists. Therefore, the sentences of “However, SIN has its associated disadvantages: a relatively short biological half-life, low bioavailability, the potential of causing clinical adverse reactions such as rash and gastrointestinal reactions by promoting histamine release, and being unstable and easily decomposable under acid, alkali, light or heat conditions [9].” were changed to “However, SIN also has some noticed characters, which include a relatively short biological half-life, the potential of causing clinical adverse reactions such as rash and gastrointestinal reactions by promoting histamine release, and being unstable and easily decomposable under acid, alkali, light or heat conditions [9].”
Point 2: Line 39, 40, 41 the sentence needs to be revised to be scientific writing.
Response 2: Line 39, 40, 41(ijms-406275 - original) has been changed to “In recent years, the anti-tumor effect of SIN has drawn worldwide attention [6-8]. However, SIN also has some noticed characters, which include a relatively short biological half-life, …”.
Point 3: One suggestion for the authors is that please show the stereochemistry of the structures in Figure 1.
Response 3: Figure 1. Stereochemistry structure of sinomenine (a), morphine (b) and codeine (c).
Thank you for your suggestion.
Best regards.